# 3D N-heterocyclic covalent organic frameworks for urea photosynthesis from $NH_3$ and $CO_2$

Ning Li [1], Jiale Zhang[1], Xiangdong Xie[2], Kang Wang [1] ✉, Dongdong Qi [1] ✉, Jiang Liu [2], Ya-Qian Lan [2] & Jianzhuang Jiang [1] ✉

Artificial photosynthesis of urea from $NH_3$ and $CO_2$ seems to remain still essentially unexplored. Herein, three isomorphic three-dimensional covalent organic frameworks with twofold interpenetrated ffc topology are functionalized by benzene, pyrazine, and tetrazine active moieties, respectively. A series of experiment results disclose the gradually enhanced conductivity, light-harvesting capacity, photogenerated carrier separation efficiency, and co-adsorption capacity towards $NH_3$ and $CO_2$ in the order of benzene-, pyrazine-, and tetrazine-containing framework. This in turn endows tetrazine-containing framework with superior photocatalytic activity towards urea production from $NH_3$ and $CO_2$ with the yield of 523 μmol $g^{-1}$ $h^{-1}$, 40 and 4 times higher than that for benzene- and pyrazine-containing framework, respectively, indicating the heterocyclic N microenvironment-dependent catalytic performance for these three photocatalysts. This is further confirmed by in-situ spectroscopic characterization and density functional theory calculations. This work lays a way towards sustainable photosynthesis of urea.

Urea serves as the most critical nitrogen fertilizer to meet the needs of a growing population[1–4]. Currently, industrial urea production is still dependent on the Bosch-Meiser two-step synthesis reaction consuming $NH_3$ and $CO_2$ under high temperature and pressure[5–9]. This complicated urea production process is energy-intensive, consuming *ca.* 30 billion kJ of energy and emitting *ca.* 2 tons of $CO_2$ for every ton of urea[10]. As a result, adapting the core synthesis chemical reactions to achieve efficient urea production under mild conditions with less energy requirement and environmentally friendliness becomes a promising alternative to conventional ones[11]. Photocatalytic technologies driven by solar energy are expected to show great potential for energy-saving production of zero-carbon urea via $CO_2$ coupled with nitrogenous substances[12,13]. Quite recently, few studies have been reported on urea photosynthesis with $N_2$ and $CO_2$ as starting material and $H_2O$ as a hydrogen source catalyzed by

inorganic nano-catalysts[14–18]. In 2022, Zhang and colleagues carried out the photosynthesis of urea with a rate of 7.2 μmol $g^{-1}$ $h^{-1}$ through the photocatalytic reduction of $CO_2$ and $N_2$ by Cu SA-$TiO_2$[14]. In 2023, urea photosynthesis was achieved from $N_2$ and $CO_2$ with yields of 6.4 and 9.2 μmol $g^{-1}$ $h^{-1}$ depending on $CeO_2$-Vo and Pd–$CeO_2$, respectively[15,16]. In the following 2024, urea production was showcased at a rate of 24.95 μmol $g^{-1}$ $h^{-1}$ under the photocatalysis of Ru particles on $TiO_2$ with $N_2/NO_3^-$ and $CO_2$ as starting materials[17]. This was followed by the photo-production of urea from $CO_2$ and $N_2$ with a rate of 78 μmol $h^{-1}$ $L^{-1}$ catalyzed by $Ni_1$-CdS/$WO_3$ in the same year[18]. However, the photocatalytic performance is unsatisfactory in terms of both the low urea production rate and in particular poor selectivity as clearly revealed in these reports. Moreover, these inorganic nano-catalysts employed also suffer from not well-defined chemical composition and structure/electronic structure, making it hard to

[1]Beijing Key Laboratory for Science and Application of Functional Molecular and Crystalline Materials, Department of Chemistry and Chemical Engineering, School of Chemistry and Biological Engineering, University of Science and Technology Beijing, Beijing, China. [2]Guangdong Provincial Key Laboratory of Carbon Dioxide Resource Utilization, School of Chemistry, South China Normal University, Guangzhou, China. ✉e-mail: kangwang@ustb.edu.cn; qdd@ustb.edu.cn; jianzhuang@ustb.edu.cn

effectively clarify the structure-activity relationship towards further improving the photocatalytic performance.

Nevertheless, it is worth noting that urea synthesis from $N_2$ and $CO_2$ starting materials and $H_2O$ as hydrogen sources under either thermal, electrochemical, or photochemical catalysis has to face the very high activation energy of 945 kJ mol$^{-1}$ for the triple N≡N bond and the very high reaction enthalpy change ($\Delta_r H_m^{\ominus}$) of 1264 kJ mol$^{-1}$ for

$$2N_2 + 2CO_2 + 4H_2O = 2CO(NH_2)_2 + 3O_2 \qquad (1)$$

In contrast, with the employment of $NH_3$ and $CO_2$ as starting materials based on the present industrial urea production route, the C-N coupling reaction for the urea generation would become a thermodynamically spontaneous one with a negative $\Delta_r H_m^{\ominus}$ of −135.5 kJ mol$^{-1}$ for[19–21]

$$2NH_3 + CO_2 = CO(NH_2)_2 + H_2O \qquad (2)$$

Furthermore, compared with the stable N≡N bond in $N_2$, the lone pair electrons in $NH_3$ are naturally reactive centers, in favor of fast reaction dynamics. As a total result, photosynthesis of urea from $NH_3$ and $CO_2$ appears more promising for future practical industrial applications. This, however, seems to remain still essentially unexplored, limited to the very late trial over $Pd@TiO_2/Gr$ catalyzed urea photosynthesis with a series of nitrogenous sources including $NO_3^-$, $N_2$, and $NH_3$ as the sole report[22], to the best of our knowledge.

Covalent organic frameworks (COFs) represent a new class of porous materials constructed from molecular building blocks linked via covalent bonds, which have shown great application potential in gas storage and separation[23], energy storage[24], optoelectronic devices[25], and catalysis owing to their advantages of low density, excellent stability, and high porosity[26–30]. In particular, their advantageous light absorption and emission characteristics, exceptional crystallinity, structural multiplicity, and stability render COFs an alluring option for potential applications in the domain of photocatalysis[31,32]. Furthermore, COFs can be meticulously modulated to create specific reaction sites and band structures through the rational design of the molecular building blocks, offering a convincing platform to verify the intrinsic mechanism and in turn achieve the desired photocatalytic performance. Recently, the incorporation of nitrogen (N)-heterocycles into COFs has been demonstrated to be a promising strategy to promote various unimolecular reactions including oxygen reduction reaction and $CO_2$ reduction reaction, due to the exemplary optical and electrical properties of N-heterocycles ascribed to their conjugated π-systems and excellent electron affinity[33,34]. Particularly, the microenvironment of catalytic reaction can be adjusted by changing the number and position of N atoms in the N-heterocycles of COFs, favoring to optimization of photocatalytic performance towards coupling reaction. In particular, N-heterocycles with directly connected N atoms such as tetrazine rings are supposed to be able to synergistically activate nitrogen and carbon sources to promote the formation of C−N coupling products.

Herein, three isomorphic three-dimensional (3D) COFs with two-fold interpenetrated ffc topology (namely 3D-TPT-COF, 3D-PDDP-COF, and 3D-TBBD-COF) were functionalized by benzene, pyrazine, and tetrazine active cores, respectively, to modulate the catalytic microenvironment through the change in the number of heterocyclic N atoms on the active cores. Electric conductivity measurement, UV-vis diffuse reflectance spectra, Nyquist plots, and transient photocurrent spectroscopic results unveil the superior semiconductivity, light absorption, and photogenerated carrier separation efficiency of 3D-TBBD-COF compared to 3D-TPT-COF and 3D-PDDP-COF owing to the N-rich nature of the tetrazine moieties in 3D-TBBD-COF. Moreover, gas adsorption experiments reveal the higher adsorption capacity of 3D-TBBD-COF to both $NH_3$ (131 cm$^3$g$^{-1}$) and $CO_2$ (33 cm$^3$g$^{-1}$) at room

temperature compared to 3D-TPT-COF (86 and 24 cm$^3$g$^{-1}$) and 3D-PDDP-COF (101 and 30 cm$^3$g$^{-1}$) owing to the more exposed lone pair electrons in the tetrazine-containing framework of 3D-TBBD-COF over the benzene-containing 3D-TPT-COF and pyridine-containing 3D-PDDP-COF. The geometrical and electronic structural advantages endow 3D-TBBD-COF with superior photocatalytic activity towards urea production from $CO_2$ and $NH_3$ with the yield of 523 μmol g$^{-1}$ h$^{-1}$, 40 and 4 times higher than that for 3D-TPT-COF and 3D-PDDP-COF, respectively, indicating the heterocyclic N microenvironment-dependent catalytic performance for these COFs photocatalysts. In addition, a series of in-situ spectroscopic characterization and density functional theory calculations demonstrate the effective synergistic co-adsorption and co-activation of $NH_3$ and $CO_2$ by the two directly connected N atoms of the tetrazine moieties in 3D-TBBD-COF due to the formation of N=N unit-based cycled intermediate depending on hydrogen bonding interaction between $NH_4^+$ and $CO_2$ dissolved in water, promoting the C-N coupling reaction to form the crucial intermediate *$NH_2C^+O$ and in turn the photocatalytic production of urea. This work lays a way towards sustainable photosynthesis of urea.

## Results and discussion

As shown in Fig. 1a and Supplementary Figs. 1–5, the solvothermal reaction of 1,3,5-tris(p-formylphenyl)benzene (TFPB) with 1,1':4',1''-terphenyl-3,3'',5,5''-tetraamine (TPT), 1,3,5-tris(4-aminophenyl)benzene (TAPB) with 5,5'-(pyrazine-2,5-diyl)diisophthalaldehyde (PDDP), and TFPB with 5,5'-(1,2,4,5-tetrazine-3,6-diyl)bis-benzene-1,3-diamine (TBBD) under the same condition generates 3D-TPT-COF, 3D-PDDP-COF, and 3D-TBBD-COF, respectively. It is worth noting that the three COFs are constructed from square-planar and trigonal-planar building units via a [4 + 3] condensation reaction, which can generate high symmetry 3D COFs with relatively high pore volume and interconnected nanochannels, favoring the exposure of the active sites and mass transfer[35]. The Fourier transform infrared (FT-IR) spectra of these three COFs show a strong C=N stretching vibration band at 1693 cm$^{-1}$, Fig. 1b. This, together with the disappearance of the C=O bond at 1724 cm$^{-1}$ and the N-H bond at 3330 cm$^{-1}$, indicates the successful condensation between the amine and aldehyde precursors[36,37]. In addition, the characteristic C signal of the imine bond is observed at 151 ppm in the $^{13}$C solid-state nuclear magnetic resonance ($^{13}$C ssNMR) spectra of the three COFs, confirming the formation of imine bonds in these COFs, Fig. 1c and Supplementary Figs. 6 and 7. Nitrogen adsorption/desorption isotherms of these COFs were obtained at 77 K to investigate their pore structure, Fig. 1d. Based on the Brunauer−Emmett−Teller (BET) model, the specific surface areas of 3D-TPT-COF, 3D-PDDP-COF, and 3D-TBBD-COF amount to 602, 461, and 556 cm$^3$g$^{-1}$, respectively, with the pore size calculated to center at 10.5, 11.1, and 11.3 Å according to the nonlocal density functional theory (NLDFT), Supplementary Fig. 8, revealing their micropore nature. It is worth noting that the inclination of isotherms with open hysteresis loops in the P/P$_0$ range of 0.2-1.0 suggests the presence of narrow and rigid slit pores between the COFs particles. Thermogravimetric analysis (TGA) discloses a decomposition temperature of *ca.* 400 °C for the three COFs, proving their high thermal stability, Supplementary Figs. 9–11. In particular, the PXRD patterns of these three COFs remain almost unchanged after being dispersed in solvents including $H_2O$, acetonitrile, ethanol, and ammonia for 7 days, Supplementary Figs. 12–14, further confirming their high stability.

The crystalline structures of 3D-TPT-COF, 3D-PDDP-COF, and 3D-TBBD-COF were determined by PXRD measurement and simulation calculation. All three COFs show a series of diffraction peaks in their PXRD patterns, Fig. 1e–g, demonstrating their good crystalline nature. As shown in Supplementary Figs. 15–24, to determine their lattice packing, 8 possible nets including 3D pto, tbo, mhq-z, fjh, gee, ffc, and 2D bex and tth topologies were constructed and optimized[35,38–40]. As can be found, the experimentally revealed PXRD pattern of 3D-TBBD-

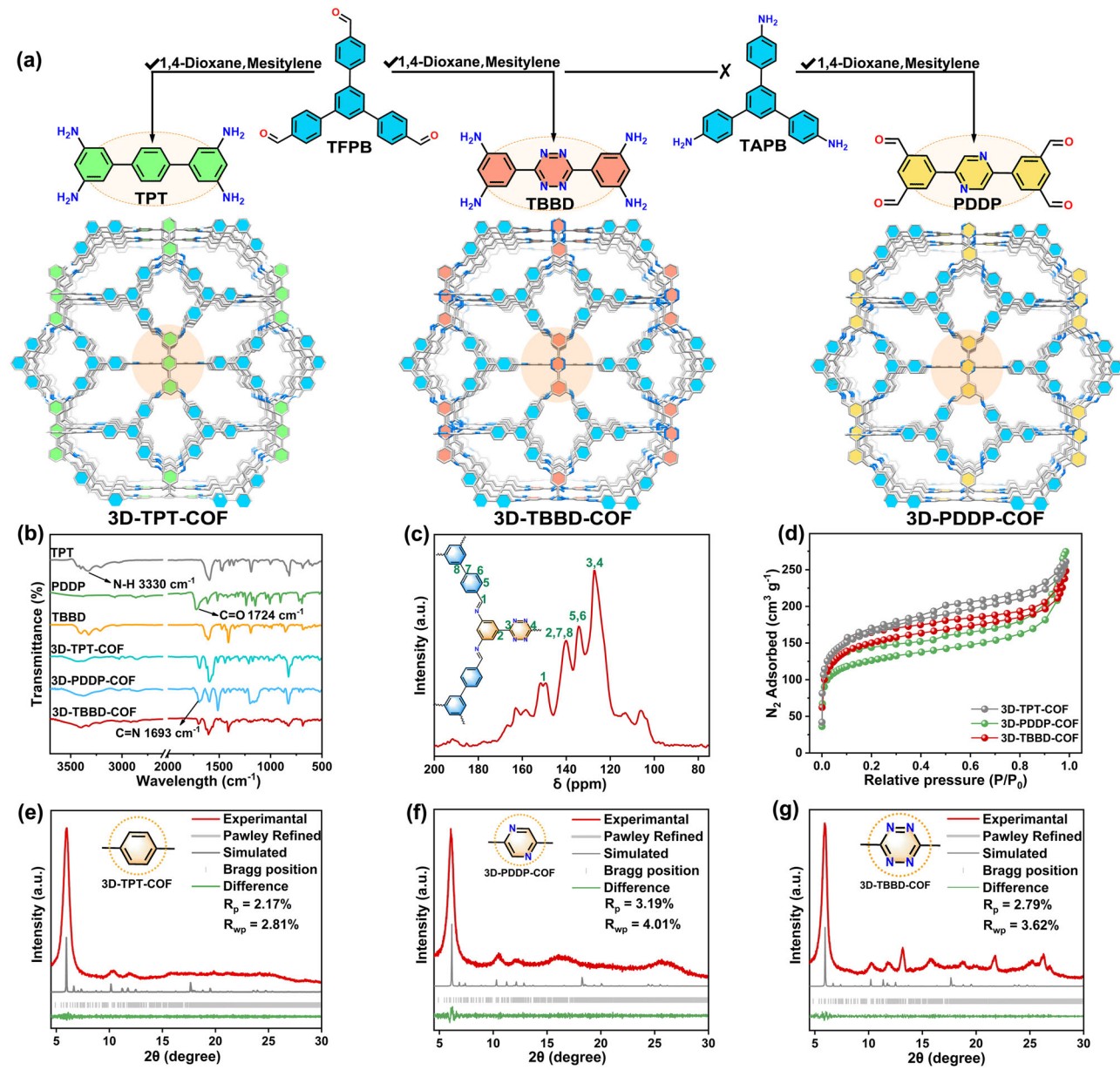

**Fig. 1 | Structural schematic and characterization of COFs. a** Schematic synthesis and (**b**) FT-IR spectra of 3D-TPT-COF, 3D-PDDP-COF, and 3D-TBBD-COF. **c** $^{13}$C ssNMR spectrum of 3D-TBBD-COF. **d** $N_2$ sorption isotherms of 3D-TPT-COF (gray), 3D-PDDP-COF (green), and 3D-TBBD-COF (red). PXRD patterns of (**e**) 3D-TPT-COF, **f** 3D-PDDP-COF, and (**g**) 3D-TBBD-COF.

COF matches well with the simulated one of the twofold interpenetrated ffc topology with the best agreement factors of $R_p$ = 2.79% and $R_{wp}$ = 3.62% according to the Pawley refinement, Fig. 1a–g. The unit cell parameters of 3D-TBBD-COF are yielded as $a$ = 82.09 Å, $b$ = 24.66 Å, $c$ = 14.58 Å, $\alpha = \beta = \gamma = 90°$ in the $P$m space group. According to the Pawley refinement, 3D-PDDP-COF and 3D-TPT-COF are isomorphic to 3D-TBBD-COF and also adopt the twofold interpenetrated ffc topology. The diffraction peaks of 3D-TBBD-COF at 5.9, 10.2, 11.9, 13.2, 18.7, and 25.1° are attributed to the (101), (421), (430), (402), (103), and (204) facets, respectively. The unit cell parameters of 3D-PDDP-COF are determined to be $a$ = 83.34 Å, $b$ = 23.61 Å, $c$ = 14.59 Å, $\alpha = \beta = \gamma = 90°$ in the $P$m space group with the observed diffraction peaks at 6.0, 10.4, 13.3, and 25.7° assigned to the (101), (421), (402), and (204) Bragg peaks, Fig. 1f. As for 3D-TPT-COF, the diffraction peaks at 5.9, 10.3, and 12.0° can be attributed to the (101), (421), and (402) facets of a lattice in the $P$m space group with cell parameters of $a$ = 83.70 Å,

$b$ = 23.47 Å, $c$ = 15.08 Å, $\alpha = \beta = \gamma = 90°$, Fig. 1e. The scanning electron microscope (SEM), transmission electron microscopy (TEM), and energy dispersive spectroscopy (EDS) were used to analyze the morphology and elemental distribution of these three 3D COFs. SEM and TEM images disclose their irregular spherical shapes with a size of *ca.* 300-500 nm, Supplementary Fig. 25. High-resolution TEM (HR-TEM) images exhibit distinct lattice stripes of 3D-TPT-COF, 3D-PDDP-COF, and 3D-TBBD-COF with a spacing of 0.98, 1.09, and 1.11 nm, respectively, Supplementary Fig. 26, corresponding to their (101) Bragg peaks, further proving their good crystalline nature. In addition, EDS mapping images reveal the uniformly distributed C and N over the COFs samples, Supplementary Fig. 27.

The semiconductivity property, light-harvesting capacity, and electronic structure of the series of three COFs were investigated by electric conductivity measurement, UV-vis diffuse reflectance spectra (UV-vis-DRS), and Mott-Schottky curves. As shown in Fig. 2a, along with

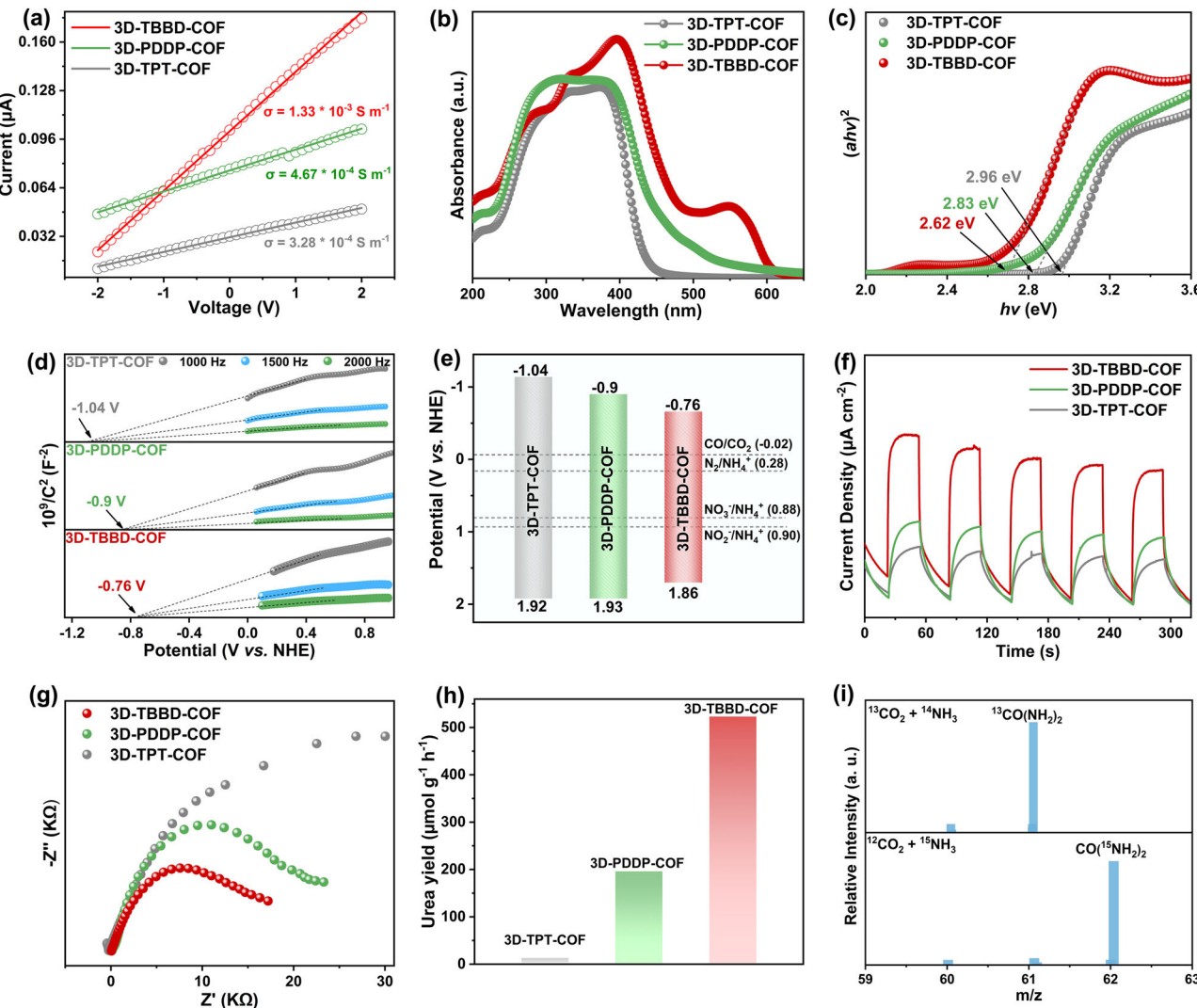

**Fig. 2 | Characterization of COFs and urea photoproduction. a** Current-voltage (**b**) UV-vis DRS, **c** Tauc plots, **d** Mott–Schottky plots, **e** energy band structure, **f** transient photocurrent spectra, and (**g**) Nyquist plots of 3D-TPT-COF (gray), 3D-PDDP-COF (green), and 3D-TBBD-COF (red). **h** Urea formation rate using 3D-TPT-COF, 3D-PDDP-COF, and 3D-TBBD-COF as photocatalyst. **i** HR-MS spectra of 3D-TBBD-COF photocatalytic $^{13}CO_2 + {}^{14}NH_3$ and $^{12}CO_2 + {}^{15}NH_3$ synthesis systems.

the replacement of the benzene moiety in 3D-TPT-COF by pyrazine moiety in 3D-PDDP-COF and tetrazine moiety in 3D-TBBD-COF, the electric conductivity gets increased from $3.28 \times 10^{-4}$ to $4.67 \times 10^{-4}$ and $1.33 \times 10^{-3}$ S m$^{-1}$ in the same order due to the electron-withdrawing inductive effect of heterocyclic N atoms[41], confirming their semi-conductive nature. Also due to the same reason, the electronic absorption range increases from 200-440 to 200-500 and 200-600 nm for 3D-TPT-COF, 3D-PDDP-COF, and 3D-TBBD-COF, respectively, Fig. 2b, indicating the improved light harvesting capacity along with the increase in the number of heterocyclic N atoms in the framework. Additionally, the bandgap widths of 3D-TPT-COF, 3D-PDDP-COF, and 3D-TBBD-COF are calculated to be 2.96, 2.83, and 2.58 eV, respectively, according to Tauc plots, Fig. 2c, revealing the narrowest band gap of 3D-TBBD-COF, which can help facilitate the transfer of photogenerated charges. Mott-Schottky measurements show a flat band of −1.04, −0.90, and −0.76 V vs. normal hydrogen electrode (NHE) for 3D-TPT-COF, 3D-PDDP-COF, and 3D-TBBD-COF, respectively, Fig. 2d. As a result, the valence band (VB) is calculated to be 1.92, 1.93, and 1.86 V vs. NHE for 3D-TPT-COF, 3D-PDDP-COF and 3D-TBBD-COF. Obviously, the CBs of these three COFs are more negative than the potential of CO₂/CO, while their VBs are more positive than the potentials of $N_2/NH_4^+$, $NO_2^-/NH_4^+$, and $NO_3^-/NH_4^+$, implying the

capacity of the three COFs to simultaneously catalyze CO₂ reduction reaction and NH₃ oxidation reaction, Fig. 2e. As shown in Fig. 2f, 3D-TBBD-COF displays higher photocurrent response signals than those for 3D-TPT-COF and 3D-PDDP-COF, indicating its enhanced photo-generated electron-hole separation efficiency owing to the introduction of tetrazine moiety into the framework. Nevertheless, 3D-TBBD-COF exhibits a smaller semicircle diameter in the Nyquist curve in comparison with 3D-TPT-COF and 3D-PDDP-COF, Fig. 2g, revealing the smallest charge transfer resistance of 3D-TBBD-COF among the three COFs. These results indicated that the photocurrent response and charge transfer characteristics of the three COFs were gradually improved along with the increase in the number of N atoms in the framework due to the enhanced electron-absorbing inductive effect of heterocyclic N atoms, in favor of improving the photocatalytic performance.

The photocatalytic performance of all the prepared COFs towards the urea production from NH₃ and CO₂ was tested in a batch reactor under the irradiation of a 300 W xenon lamp. Photocatalytic tests were performed in pure water by adding NH₃ and CO₂ (2:1, v/v) as feedstock and the prepared COFs as the catalysts. As shown in Fig. 2h and Supplementary Fig. 28, 3D-TPT-COF without N-containing het-erocycles in the framework displays a very small urea production rate

of 13 µmol g$^{-1}$ h$^{-1}$, revealing its negligible photocatalytic activity towards urea production from $NH_3$ and $CO_2$. Replacing the benzene moieties in 3D-TPT-COF with the pyrazine moieties in 3D-PDDP-COF leads to a significantly increased urea production rate of 196 µmol g$^{-1}$ h$^{-1}$ for 3D-PDDP-COF, suggesting the effect of N-containing heterocycles in the framework on promoting the formation of urea from $NH_3$ and $CO_2$ coupling. Further increase in the N atomic number via replacing the pyrazine moiety in 3D-PDDP-COF with tetrazine moiety in 3D-TBBD-COF results in a further significantly increased urea production rate of 523 µmol g$^{-1}$ h$^{-1}$ for the latter material, confirming the active center nature of N atoms in the N-containing heterocycles of the frameworks towards urea production. In particular, the four times higher urea production rate of 3D-TBBD-COF in comparison with 3D-PDDP-COF also implies the synergistic effect of the two directly connected N active sites in a pair of N=N units of tetrazine moieties on achieving C-N coupling for the formation of urea from $NH_3$ and $CO_2$. Furthermore, the apparent quantum yield (AQY) of 3D-TBBD-COF was measured at different wavelengths with a maximum value of 0.32% observed at 420 nm, Supplementary Fig. 29. Impressively, the urea yield of 3D-TBBD-COF remains almost constant after five consecutive cycles of the reaction, Supplementary Fig. 30, revealing the excellent photostability of 3D-TBBD-COF. In addition, the recycled 3D-TBBD-COF sample after photocatalytic cycles exhibits a very similar PXRD pattern, XPS spectrum, and FT-IR spectrum to those for the 3D-TBBD-COF sample before photocatalytic cycles, Supplementary Figs. 31–33, further confirming its excellent stability during photocatalytic reaction process. At the end of this section, it is noteworthy that both conventional industrial production and electrochemical production can form ca. 1.4-2.0 tons of $CO_2$ per ton of urea. In contrast, utilizing the present photocatalytic process, carbon dioxide acts as a continuously consuming feedstock. In addition, the costs of energy and feedstock input for the photocatalytic urea production were also calculated based on the life cycle cost approach. According to the calculation result, photocatalytic urea production in the present case offers lower cost, USD 70/ton urea, than conventional industrial urea production, USD 410/ton urea as well as electrochemical urea production, USD 360/ton urea. With an annual production of 21 million tons of urea, this photocatalytic urea production can save 6 billion USD, in detail for the data sources see Supplementary note[10,42,43].

To identify the role of 3D-TBBD-COF in the photocatalytic system, various control experiments were carried out, Supplementary Figs. 34 and 35. In the absence of COF photocatalysts, no urea was generated in the reactor, manifesting the photocatalyst nature of 3D-TBBD-COF in photocatalytic $NH_3$ and $CO_2$ coupling reaction to produce urea. As expected, the urea production reaction carried out in the dark gives no target product, disclosing the light irradiation promotion nature for the urea production from $NH_3$ and $CO_2$. Nevertheless, no urea was detected without adding $NH_3$ or $CO_2$ into the reactor, revealing the origination of the urea from the coupling reaction between $CO_2$ and $NH_3$. To confirm the nitrogen and carbon sources of urea generated, photocatalytic experiments were performed with $^{15}NH_3$ and $^{13}CO_2$ as the feedstock, respectively. As displayed in Fig. 2i and Supplementary Fig. 36, the employment of $^{15}NH_3$ and $^{13}CO_2$ as the reactant leads to the observation of a signal at m/z = 62 and 61, respectively, in the high-resolution mass (HR-MS) spectra, corresponding to the $^{15}N$ and $^{13}C$ labeled urea molecules, confirming the urea origination from $NH_3$ and $CO_2$ coupling reaction.

In-situ diffuse reflectance infrared fourier transform (DRIFT) spectroscopy was further performed to trace the adsorption process of $CO_2$ and $NH_3$ on the COFs, Fig. 3a–c and Supplementary Figs. 37 and 38. As shown in Fig. 3a, b, in the in-situ DRIFT spectra of 3D-TBBD-COF for $CO_2$ adsorption in water, the peaks at 1668, 1542, and 1350 cm$^{-1}$ due to the C = O, COO$^-$, and HCO$_3^-$ stretching bands of the adsorbed $CO_2$ molecules appeared and increased along with the

increase of adsorption time[44,45], while new two bands get appeared at 1360, and 1030 cm$^{-1}$ due to the N-H stretching and bending vibration of the adsorbed $NH_3$ in the in-situ DRIFT spectra of 3D-TBBD-COF for $NH_3$ adsorption in water[46,47]. These results prove the $CO_2$ and $NH_3$ adsorption capacity of 3D-TBBD-COF. In particular, the C = O and COO$^-$ vibration band at 1600, and 1517 cm$^{-1}$ and N-H vibration band at 1363 and 1029 cm$^{-1}$ are more significantly observed in the in-situ DRIFT spectra of 3D-TBBD-COF for $CO_2$ and $NH_3$ co-adsorption compared to those for sole $CO_2$ or $NH_3$ adsorption, Fig. 3c, revealing synergistic adsorption of 3D-TBBD-COF for $CO_2$ and $NH_3$ owing to the spatial confinement effect induced by the tetrazine moieties as detailed below[48–50]. In comparison, 3D-TPT-COF exhibit obviously weaker bands at 1665 and 1050 cm$^{-1}$ than 3D-TBBD-COF and 3D-PDDP-COF during the absorption processes of $CO_2$ and $NH_3$, Supplementary Figs. 37 and 38, revealing the weakened adsorption capacity of 3D-TPT-COF for $CO_2$ and $NH_3$. This in turn leads to their inferior photocatalytic performance towards $NH_3$ and $CO_2$ coupling to produce urea.

The $NH_3/CO_2$ adsorption isotherm measurements were carried out at room temperature by static volumetric method to investigate the adsorption activity of the prepared three COFs towards the reaction substrates. As shown in Supplementary Figs. 39–42, the $NH_3$ and $CO_2$ uptakes of 3D-TPT-COF at P/P$_0$ = 1.0 under room temperature amount to 86 and 24 cm$^3$g$^{-1}$, respectively. As the increase in the number of heterocyclic N atoms in the framework, the $NH_3/CO_2$ adsorption capacity of 3D-PDDP-COF and 3D-TBBD-COF increased to 101/30 and 131/33 cm$^3$g$^{-1}$ at P/P$_0$ = 1.0 and 298 K, respectively. Due to their same twofold interpenetrated ffc topology and very close specific surface area, these results indicate the effect of the heterocyclic N atoms on enhancing the $NH_3$ and $CO_2$ adsorption capacity, which in turn can promote the photocatalytic $NH_3$ and $CO_2$ coupling reaction to generate urea.

To clarify the adsorption mechanism for the $CO_2$ and $NH_3$ on the COFs, adsorption energy calculations were performed at the level of M06-2X/6-311 G(d). As shown in Fig. 3d, e, once $NH_3$ is dissolved into water, its main existing form will immediately change to $NH_4^+$ ion. It is worth noting that as a typical electron-deficient moiety, tetrazine (Tz) moiety with two N=N units in 3D-TBBD-COF would certainly receive electrons from the adjacent benzene moieties. This results in a negative-charged active center on the Tz moiety, endowing Tz moiety with the adsorption capacity for $CO_2$ (also dissolved in water) and $NH_4^+$ with an adsorption energy of −85 and −13 kJ mol$^{-1}$, respectively, due to the mutual attraction of positive/negative charged centers. In particular, the very short N...N distance of ~1.4 Å in the N=N unit of tetrazine moiety allows the formation of a co-adsorbing structure of heptatomic ring with an increased adsorption energy of ~136 kJ mol$^{-1}$ due to the formation of O = C = O...H-NH$_3^+$ hydrogen bonding, favoring the C-N coupling from $CO_2$ and $NH_3$, Fig. 3e. In contrast, due to the lack of two directly connected N atoms, the same co-adsorbing and co-activation structure cannot be formed in either 3D-PDDP-COF or 3D-TPT-COF, Fig. 3d, e and Supplementary Figs. 43 and 44.

The catalytic conversion process of $CO_2$ and $NH_3$ on the COFs was also monitored by in-situ DRIFT spectroscopy, Fig. 3f–h. As expected, new absorption bands appeared at 1702, and 1405 cm$^{-1}$ in the in-situ DRIFT spectra of the 3D-TBBD-COF photocatalytic system, corresponding to the antisymmetrical *NH$_2$C$^+$O stretching vibration of urea and the C-N stretching vibration of both urea and reaction intermediates[51], respectively, Fig. 3h. Their intensity steadily grows along with the reaction time, disclosing the successful C-N coupling and the production of urea from $NH_3$ and $CO_2$ under the photocatalysis of 3D-TBBD-COF. In addition, the intensity for the band at 1052 cm$^{-1}$ due to adsorbed $NH_3$ gets significantly decreased due to the consumption of adsorbed $NH_3$ after C−N coupling. In comparison, the in-situ DRIFT spectra of the 3D-PDDP-COF photocatalytic system display very weak bands at 1401 cm$^{-1}$ even after photoreaction for 180 s,

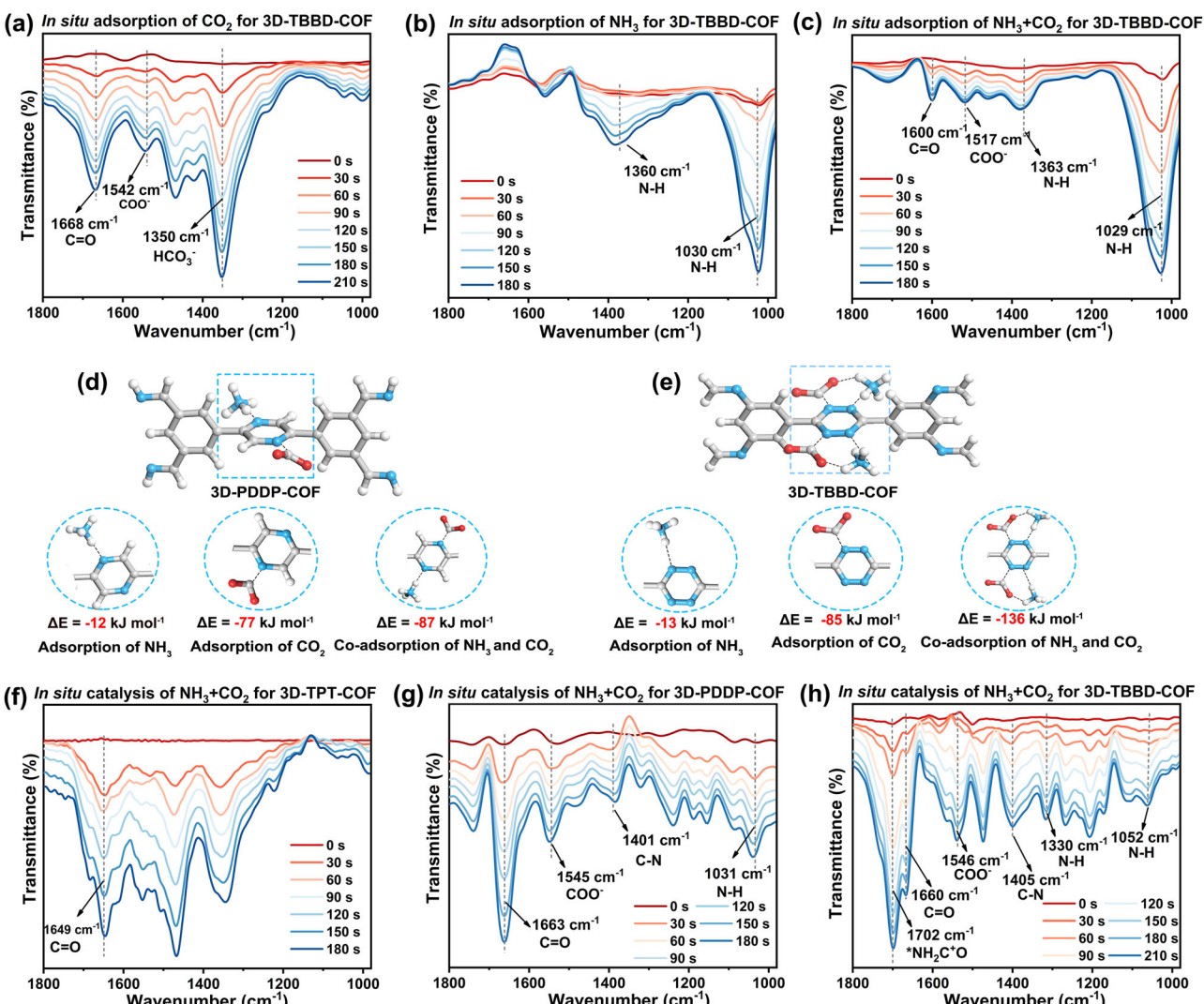

**Fig. 3 | In situ DRIFT spectra and adsorption structure of COFs.** In-situ DRIFT spectra of 3D-TBBD-COF for (**a**) $CO_2$, **b** $NH_3$, and (**c**) $CO_2 + NH_3$ adsorption at different times, respectively. Adsorption structures and adsorption energies for $CO_2$, $NH_3$, and $CO_2 + NH_3$ on (**d**) 3D-PDDP-COF and (**e**) 3D-TBBD-COF (C: gray; N: blue; O: red; H: white). In-situ DRIFT spectra of (**f**) 3D-TPT-COF, **g** 3D-PDDP-COF, and (**h**) 3D-TBBD-COF photocoupling $CO_2$ and $NH_3$ for urea production at different times.

Fig. 3g, manifesting the lower activity of 3D-PDDP-COF for C-N coupling from $CO_2$ and $NH_3$. Moreover, the absorption bands due to the C-N and antisymmetric $*NH_2C^+O$ stretching vibrations are not observed in the in-situ DRIFT spectra of the 3D-TPT-COF photocatalytic system, Fig. 3f, indicating its incapability of photocatalytic C-N coupling from $CO_2$ and $NH_3$ owing to the lack of N-containing aromatic moiety, in line with other experimental findings as mentioned above. 3D-TBBD-COF exhibits significantly enhanced $CO_2$ and $NH_3$ co-adsorption and co-activation performance due to the existence of N = N units in the tetrazine moiety of 3D-TBBD-COF. Obviously, the regulation of the microenvironment by changing the number of N atoms in the active core can effectively improve the adsorption and catalysis of COFs towards the reaction substrates.

To clarify the photocatalytic mechanism for the urea generation from $CO_2$ and $NH_3$ on the COFs, the Gibbs free energy calculations were carried out at the level of M06-2X/6-311 G(d). Based on the calculation results, the full catalytic route on 3D-TBBD-COF could be divided into three stages: (1) Photon capture and generation of excited tetrazine moiety (Tz*); (2) the first C-N coupling process with the help of electron-withdrawing inductive effect of two directly connected N atoms in the N = N units of Tz moiety; (3) the second C-N coupling

process owing to the spatial confinement effect induced by the Tz moiety. As can be seen in Fig. 4a, b, once the photosensitizer Tz captures a photon with a wavelength in the range of 200-600 nm, the reaction will start from the hydrogen transfer from $NH_4^+$ to $CO_2$ along the heptatomic ring according to Eqs. (3) and (4) with an energy barrier of 1.31 eV. The next step involves the dehydration process to produce two unstable intermediate species $*C^+O$ and $*NH_2$ with an unpaired electron according to Eq. (5). In the next moment, $*C^+O$ and $*NH_2$ will quickly combine into the half urea $NH_2C^+O$ according to Eq. (6), completing the first C-N coupling process.

$$Tz + CO_2 + NH_4^+ \rightarrow CO_2 - Tz - NH_4^+ \qquad (3)$$

$$CO_2 - Tz - NH_4^+ + h\nu \rightarrow C^+O_2H - Tz - NH_3 \qquad (4)$$

$$C^+O_2H - Tz - NH_3 \rightarrow [C^+O - Tz - NH_2]^{\neq} + H_2O \qquad (5)$$

$$C^+O - Tz - NH_2 \rightarrow NH_2C^+O - Tz \qquad (6)$$

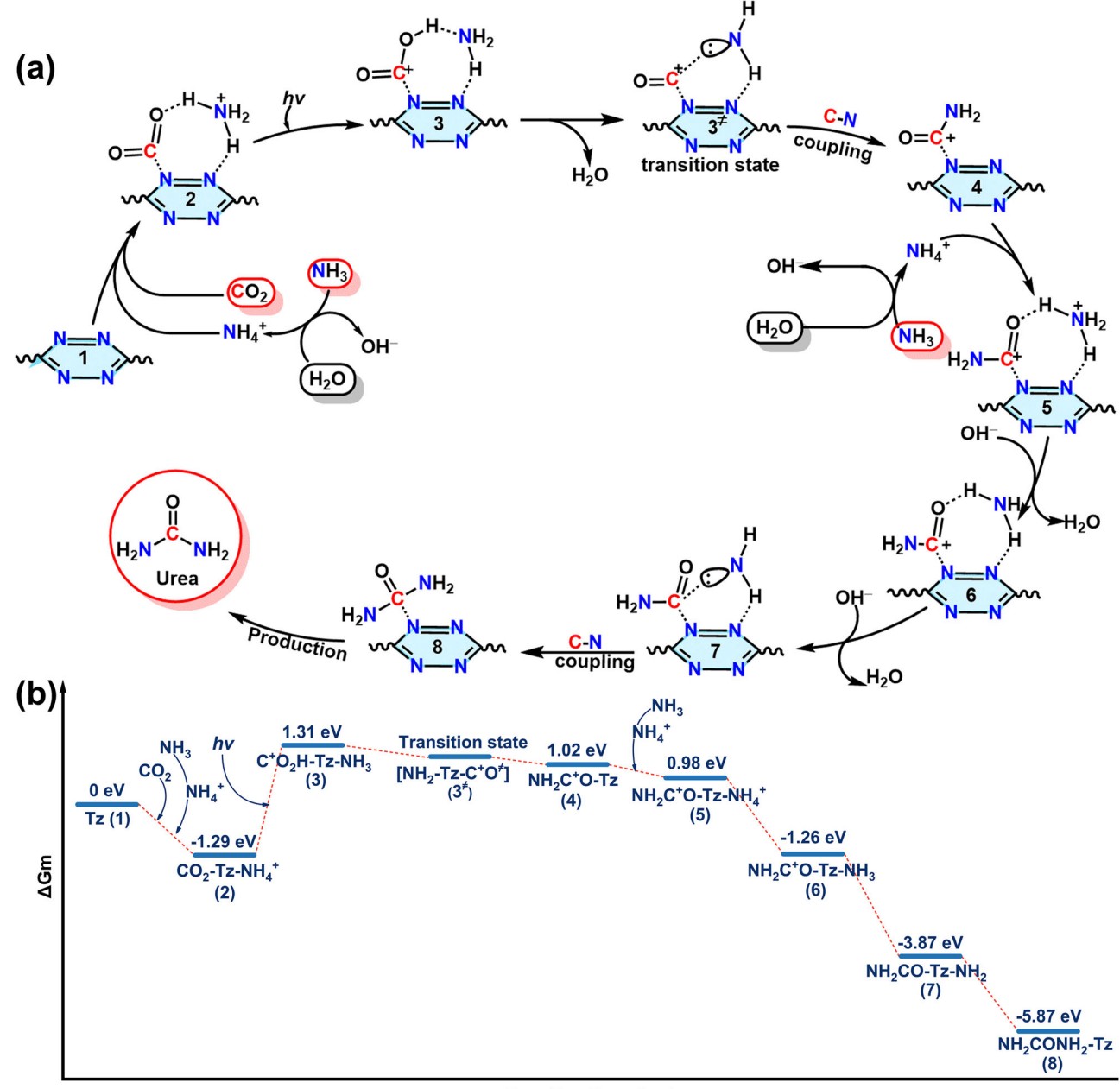

**Fig. 4 | DFT calculations. a** The reaction pathway and (**b**) corresponding free energy for the generation of urea from $CO_2$ and $NH_3$ on 3D-TBBD-COF.

After the construction of $NH_2C^+O$, one of the two directly connected N absorbing sites in the N = N unit of tetrazine moiety becomes empty again, which would attract a new $NH_4^+$ ion and form a new co-adsorbing structure of $NH_2C^+O$-Tz-$NH_4^+$ according to Eq. (7). Immediately, the dehydration (Eqs. (8) and (9)) and C-N coupling processes (Eq. (10)) are repeated in this stage, leading to the formation of the urea $NH_2CONH_2$.

$$NH_2C^+O - Tz + NH_4^+ \rightarrow NH_2C^+O - Tz - NH_4^+ \quad (7)$$

$$NH_2C^+O - Tz - NH_4^+ + OH^- \rightarrow NH_2C^+O - Tz - NH_3 \quad (8)$$

$$NH_2C^+O - Tz - NH_3 + OH^- \rightarrow NH_2C^+O - Tz - NH_2 + H_2O \quad (9)$$

$$NH_2C^+O - Tz - NH_2 \rightarrow NH_2CONH_2 - Tz \quad (10)$$

When we look back at the whole catalytic route on 3D-TBBD-COF, it is obvious that all the reactants are crowded together in a very narrow space limited by the two directly connected N atoms with a very short N····N distance of ~1.4 Å in the N = N units of tetrazine moiety, favoring the C-N coupling between *$C^+O$ and *$NH_2$ intermediates without energy barrier. This, however, is not the case for 3D-PDDP-COF and 3D-TPT-COF. For 3D-PDDP-COF, the two reactants, $CO_2$ and $NH_4^+$, can not depend on the two oppositely arranged N atoms in one pyrazine moiety with N····N distance of ~2.6 Å to proceed with the C-N coupling, Supplementary Figs. 43 and 44. Fortunately, these two reactants are able to depend on two pyrazine N atoms in neighboring layers in the framework with N····N distance of ~3.5 Å to complete the C-N coupling. This, however, requires an obvious energy barrier of 2.00 eV, leading to a significantly weakened photocatalytic $CO_2$ and $NH_3$ coupling efficiency, Supplementary Figs. 45 and 46. This is in good agreement with the experimental findings as detailed above. For 3D-TPT-COF,

$CO_2$ and $NH_3$ co-adsorption and the following C-N coupling could not be achieved owing to the lack of N adsorption sites in the benzene moiety for $CO_2$ and $NH_4^+$. As a total result, the strong spatial confinement effect due to the two directly connected N atoms in the $N = N$ units of tetrazine moiety plays a key role in the C-N coupling process in comparison with 3D-PDDP-COF and 3D-TPT-COF, leading to the efficient urea production with a high rate on 3D-TBBD-COF.

In summary, a series of 3D COFs functionalized with benzene, pyrazine, and tetrazine active cores have been designed and synthesized. The catalytic microenvironment of these COFs can be adjusted by increasing the number of heterocyclic N atoms in the framework, leading to the gradually enhanced light-harvesting capacity, photogenerated carrier separation efficiency, and co-adsorption capacity towards $NH_3$ and $CO_2$ in the order of benzene-, pyrazine-, and tetrazine-containing framework. This enables the tetrazine-containing 3D-TBBD-COF to exhibit much superior photocatalytic activity towards urea production from $CO_2$ and $NH_3$ to 3D-TPT-COF and 3D-PDDP-COF. In situ DRIFTS investigation and theoretical calculations unveil the C-N coupling reaction between $*C^+O$ and $*NH_2$ intermediates to form crucial intermediate $*NH_2C^+O$ derived from $CO_2$ reduction and $NH_3$ oxidation over the $N = N$ centers of tetrazine moiety in 3D-TBBD-COF during the urea formation process. This work lays a way towards sustainable photosynthesis of urea.

## Methods

### Synthesis of 3D-TBBD-COF
TFPB (15.6 mg, 0.04 mmol) and TBBD (8.8 mg, 0.03 mmol) were weighed into a Pyrex tube with 0.5 mL 1,4-Dioxane and 0.5 mL mesitylene. The mixture was subjected to ultrasonic treatment for 30 minutes to achieve a suspension. Subsequently, 0.2 mL of 6 M acetic acid was added, followed by another 10 minutes of ultrasonic processing. The solution then underwent three freeze-pump-thaw cycles. Thereafter, the Pyrex tubes were hermetically sealed and heated in an oven at 120 °C for 4 days. The precipitate was subsequently isolated via centrifugation and underwent purification through sequential treatments with acetone and tetrahydrofuran for 24 hours. Ultimately, 3D-TBBD-COF was then obtained as a powder with a yield of 65%.

### Synthesis of 3D-PDDP-COF
TAPB (14.1 mg, 0.04 mmol) and PDDP (10.4 mg, 0.03 mmol) were weighed into a Pyrex tube with 0.5 mL 1,4-Dioxane and 0.5 mL mesitylene. The mixture was subjected to ultrasonic treatment for 30 minutes to achieve a suspension. Subsequently, 0.2 mL of 6 M acetic acid was added, followed by another 10 minutes of ultrasonic processing. The solution then underwent three freeze-pump-thaw cycles. Thereafter, the Pyrex tubes were hermetically sealed and heated in an oven at 120 °C for 4 days. The precipitate was subsequently isolated via centrifugation and underwent purification through sequential treatments with acetone and tetrahydrofuran over 24 hours. Ultimately, 3D-PDDP-COF was then obtained as a powder with a yield of 60%.

### Synthesis of 3D-TPT-COF
TFPB (15.6 mg, 0.04 mmol) and TPT (8.7 mg, 0.03 mmol) were weighed into a Pyrex tube with 0.5 mL 1,4-Dioxane and 0.5 mL mesitylene. The mixture was subjected to ultrasonic treatment for 30 minutes to achieve a suspension. Subsequently, 0.2 mL of 6 M acetic acid was added, followed by another 10 minutes of ultrasonic processing. The solution then underwent three freeze-pump-thaw cycles. Thereafter, the Pyrex tubes were hermetically sealed and heated in an oven at 120 °C for 4 days. The precipitate was subsequently isolated via centrifugation and underwent purification through sequential treatments with acetone and tetrahydrofuran over

24 hours. Ultimately, 3D-TPT-COF was then obtained as a powder with a yield of 80%.

## Data availability
All relevant data that support the findings of this study are presented in the manuscript and supplementary information file. Source data are provided with this paper.

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

## Acknowledgements

Financial support from the Natural Science Foundation of China, grant No. 22235001 (J.J.) and 22175020 (J.J.).

## Author contributions

Conceptualization: J.J., K.W., N.L. Methodology: K.W., N.L. Investigation: N.L., J.Z., X.X. Visualization: K.W., N.L., J.L., Y.L. DFT Calculation: D.Q., N.L. Writing—original draft: J.J., K.W., N.L. Writing—review and editing: J.J., K.W., N.L., D.Q.

## Competing interests

The authors declare no competing interests.
