## [Peer Review File · Nature Communications]

3D N-heterocyclic Covalent Organic Frameworks for Urea Photosynthesis from NH₃ and CO₂

Corresponding Author: Professor Jianzhuang Jiang

Version 0:

Reviewer comments:

Reviewer #1

(Remarks to the Author)

The manuscript of "Three-dimensional Tetrazine-containing Covalent Organic Framework for Urea Photosynthesis" presents the synthesis and photocatalytic urea production of three isostructural 3D COFs. These COFs have been deduced from the condensation between either tetraamine and trialdehyde or tetraaldehyde and triamine. In particular, 3D-TBBD-COF displays a significant photocatalytic property with the outstanding urea [CO(NH₂)₂] rate of 523 μmol g⁻¹ h⁻¹. Moreover, the in-situ spectroscopic characterization and density functional theory calculations disclose the synergistic catalytic mechanism. The present work is suitable for Nat. Commun. due to its significance in promoting photosynthesis of urea from NH₃ and CO₂ with convincing production yield for the first time. However, the following issues should be addressed before the final acceptance.

- (1) Figure 1a seems to be wrong, because TAPB and TBBD both with amino groups do not react.
- (2) TBBD is very sensitive to light. Please test the photocatalytic product only in present sole CO₂ or NH₃.
- (3) The authors claim that "...reveal the strong adsorption of 3D-TBBD-COF to both NH₃ and CO₂ with high adsorption capacity of 131 and 33 cm³ g⁻¹ at room temperature". While the CO₂ uptake is not high. Please correct the sentence.
- (4) According to the previous results, the imine orientation plays an important role in photocatalysis. As described in the text, 3D-PDDP-COF and 3D-TBBD-COF with different the imine orientation and active units have the different photocatalytic urea production rate. Does imine orientation play a critical effect in the present photosynthesis?
- (5) There is several format errors. For example, in P/Po, "o" should be corrected as "0". Please further check the manuscript.
- (6) Please try to assign the ¹³C ssNMR spectrum of COFs, although it is very difficult.
- (7) The experimental pore sizes should be compared with that of simulated structural models.
- (8) The detailed synthesis of 5,5'-(pyrazine-2,5-diyl) diisophthalaldehyde (PDDP) and 5,5'-(1,2,4,5-tetrazine-3,6-diyl) bis-benzene-1,3-diamine (TBBD) should be added in SI.

Reviewer #2

(Remarks to the Author)

Photocatalytic technologies have great potential for energy saving production of zero-carbon urea. In particular, urea photosynthesis from NH₃ and CO₂ appears more promising towards future practical industrial application compared to that from N₂ and CO₂, which however has rarely been unexplored. In this paper, three isomorphous 3D COFs were functionalized by benzene, pyrazine, and tetrazine active moieties to modulate the catalytic microenvironment through altering the number of heterocyclic N atoms on the active cores. It has been revealed that the conductivity, light harvesting capacity, photogenerated carrier separation efficiency, and co-adsorption capacity towards NH₃ and CO₂ get gradually enhanced in

the order of benzene-, pyrazine-, and tetrazine-containing framework. This endows the prepared tetrazine-containing COF with superior photocatalytic activity towards urea production from CO₂ and NH₃ with a high yield of 523 μmol g⁻¹ h⁻¹. Moreover, the catalytic mechanism was also demonstrated by in-situ spectroscopic characterization and density functional theory calculation. The present result is of significance towards the exploration of sustainable urea photosynthesis and the rational design of functional COFs. In addition, this work is well organized. As a result, I recommend this manuscript for publication after minor revision.

- (1) The authors should highlight the novelty of urea photosynthesis from NH₃ and CO₂ by use of tetrazine-containing COF.
- (2) Figure 1b is not clear, which is suggested to be redrawn.
- (3) The bandgap calculated from Figure 2e is not consistent with those shown in Figure 2d.
- (4) The unit of the ordinate in Figure 2e should be "V vs. NHE" rather than "eV", while the unit of photocurrent was missing in Figure 2f.
- (5) Some acronyms such as PXRD and UV-vis-DRS should be defined at their first usage.
- (6) The authors claimed "This is also supported by the almost unchanged PXRD pattern and XPS spectrum of 3D-TBBD-COF after photocatalytic cycles, Figures S30 and S31". However, Figures S30 and S31 show that the spectra after cycling are not the same with from those before cycling. The authors should comment on this phenomenon.
- (7) For comparison, the adsorption structures and adsorption energy for CO₂, NH₃, and CO₂+NH₃ on 3D-TPT-COF should be also provided.

Reviewer #3

(Remarks to the Author)

This is an interesting manuscript on the topic of Covalent Organic Frameworks for Urea Photosynthesis from NH₃ and CO₂. The manuscript is well-written, and the logic behind the work is sound. Therefore, I would like to recommend acceptance with minor revisions.

The COFs exhibit modest porosity. Is this the reason for the overall moderate performance? Additionally, the rationale behind selecting these specific COFs should be elaborated upon.

Many COFs have been reported for use in urea synthesis. How does the performance of these COFs compare to those previously reported? Furthermore, the authors should provide details on the chemical stability of the COFs before and after catalysis.

Version 1:

Reviewer comments:

Reviewer #1

(Remarks to the Author)

The manuscript is now recommended for acceptance.

Reviewer #2

(Remarks to the Author)

After the revision, this work could be accepted.

Reviewer #3

(Remarks to the Author)

The authors have done an excellent job, and this manuscript could be accepted as it is.

Response to reviewers' reports:

Reviewer 1:

Comment: *The manuscript of “Three-dimensional Tetrazine-containing Covalent Organic Framework for Urea Photosynthesis” presents the synthesis and photocatalytic urea production of three isostructural 3D COFs. These COFs have been deduced from the condensation between either tetraamine and trialdehyde or tetraaldehyde and triamine. In particular, 3D-TBBD-COF displays a significant photocatalytic property with the outstanding urea [CO(NH₂)₂] rate of 523 μmol g⁻¹ h⁻¹. Moreover, the in-situ spectroscopic characterization and density functional theory calculations disclose the synergistic catalytic mechanism. The present work is suitable for Nat. Commun. due to its significance in promoting photosynthesis of urea from NH₃ and CO₂ with convincing production yield for the first time. However, the following issues should be addressed before the final acceptance.*

Answer: First of all, we really appreciate this reviewer's affirmation regarding our work and the valuable suggestions towards improving the quality of the present research. The responses to the specific comments are detailed below.

1. *Figure 1a seems to be wrong, because TAPB and TBBD both with amino groups do not react.*

Answer: Thanks a lot for pointing out this error. We truly apologize for having made such kind of mistake. Accordingly, **Figure 1a** has been modified in the revised version of manuscript. In addition, we have also tried our best to check throughout the whole manuscript to avoid such kind of mistakes.

2. *TBBD is very sensitive to light. Please test the photocatalytic product only in present sole CO₂ or NH₃.*

Answer: Thanks a lot for this suggestion. Accordingly, photocatalytic tests for the TBBD in the presence of sole CO₂ or NH₃ have been performed. As shown in **Figure S35** in the revised version of Supporting Information (actually **Figure R4** also given below), CO with a yield of 110 μmol g⁻¹ was determined in the presence of sole CO₂ while NO₂⁻ with a yield of 315 μmol g⁻¹ was found in the presence of sole NH₃, demonstrating the photocatalytic activity of TBBD towards CO₂ reduction and NH₃ oxidation.

Figure R4. The photocatalytic product of 3D-TBBD-COF in the presence of sole CO₂ and sole NH₃.

3. The authors claim that “...reveal the strong adsorption of 3D-TBBD-COF to both NH₃ and CO₂ with high adsorption capacity of 131 and 33 cm³ g⁻¹ at room temperature”. While the CO₂ uptake is not high. Please correct the sentence.

Answer: Thanks a lot for this suggestion. Accordingly, “...reveal the strong adsorption of 3D-TBBD-COF to both NH₃ and CO₂ with high adsorption capacity of 131 and 33 cm³ g⁻¹ at room temperature” has been changed into “gas adsorption experiments reveal the higher adsorption capacity of 3D-TBBD-COF to both NH₃ (131 cm³ g⁻¹) and CO₂ (33 cm³ g⁻¹) at room temperature compared to 3D-TPT-COF (86 and 24 cm³ g⁻¹) and 3D-PDDP-COF (101 and 30 cm³ g⁻¹).....”

4. According the previous results, the imine orientation plays an important role in photocatalysis. As described in the text, 3D-PDDP-COF and 3D-TBBD-COF with different the imine orientation and active units have the different photocatalytic urea production rate. Does imine orientation play a critical effect in the present photosynthesis?

Answer: Thanks a lot for this question. Indeed, 3D-PDDP-COF has different imine orientation from 3D-TPT-COF and 3D-TBBD-COF. However, 3D-PDDP-COF exhibits a photocatalytic urea production rate of 196 μmol g⁻¹ h⁻¹, significantly lower than that for 3D-TBBD-COF (523 μmol g⁻¹ h⁻¹) but much higher than that for 3D-TPT-COF (13 μmol g⁻¹ h⁻¹), suggesting the insignificant effect of the imine orientation on the photocatalytic activity of these three COFs.

5. There is several format errors. For example, in P/Po, “o” should be corrected as “0”. Please further check the manuscript.

Answer: Thanks a lot for pointing out our typewriting mistakes. We feel very sorry for having made such kind of mistakes. Accordingly, “P/Po” has been changed into “P/P₀” in the revised version of manuscript. In addition, we have also tried our best to check throughout the whole manuscript to avoid such kind of typewriting mistakes.

6. Please try to assign the ^{13}C ssNMR spectrum of COFs, although it is very difficult.

Answer: Thanks a lot for this suggestion. Accordingly, we tried to assign the ^{13}C ssNMR spectra of all the three COFs, **Figures 1c, S6, and S7** in the revised version of manuscript and Supporting Information (actually **Figure R5** also given below).

Figure R5. ^{13}C ssNMR spectrum of (a) 3D-TPT-COF, (b) 3D-PDDP-COF and (c) 3D-TBBD-COF.

7. The experimental pore sizes should be compared with that of simulated structural models.

Answer: Thanks a lot for this suggestion. On the basis of the N_2 adsorption test results, the pore sizes of 3D-TPT-COF, 3D-PDDP-COF, and 3D-TBBD-COF amount to 10.5, 11.1, and 11.3 Å, respectively, in agreement with the result deduced from the simulated structural models, 11.4 Å for 3D-TPT-COF, 11.3 Å for 3D-PDDP-COF, and 11.4 Å for 3D-TBBD-COF.

8. The detailed synthesis of 5,5'-(pyrazine-2,5-diyl) diisophthalaldehyde (PDDP) and 5,5'-(1,2,4,5-tetrazine-3,6-diyl) bis-benzene-1,3-diamine (TBBD) should be added in SI.

Answer: Thanks a lot for this instruction. Accordingly, the detailed synthesis of PDDP and TBBD has been provided in the revised version of Supporting Information.

Reviewer: 2

Comment: Photocatalytic technologies have great potential for energy saving production of zero-carbon urea. In particular, urea photosynthesis from NH_3 and CO_2 appears more promising towards future practical industrial application compared to that from N_2 and CO_2 , which however has rarely been unexplored. In this paper, three isomorphic 3D COFs were functionalized by benzene, pyrazine, and tetrazine active moieties to modulate the catalytic microenvironment through altering the number of heterocyclic N atoms on the active cores. It has been revealed that the conductivity, light harvesting capacity, photogenerated carrier separation efficiency, and co-adsorption capacity towards NH_3 and CO_2 get gradually enhanced in the order of

benzene-, pyrazine-, and tetrazine-containing framework. This endows the prepared tetrazine-containing COF with superior photocatalytic activity towards urea production from CO₂ and NH₃ with a high yield of 523 μmol g⁻¹ h⁻¹. Moreover, the catalytic mechanism was also demonstrated by in-situ spectroscopic characterization and density functional theory calculation. The present result is of significance towards the exploration of sustainable urea photosynthesis and the rational design of functional COFs. In addition, this work is well organized. As a result, I recommend this manuscript for publication after minor revision.

Answer: Thanks a lot for the reviewer's precious comments to enhance the quality of our work. The responses to the specific comments are detailed below.

1. *The authors should highlight the novelty of urea photosynthesis from NH₃ and CO₂ by use of tetrazine-containing COF.*

Answer: Thanks a lot for this suggestion. It is worth noting that the microenvironment of catalytic reaction can be adjusted by changing the number and position of N atoms in the N-heterocycles of COFs, favoring to optimize photocatalytic performance towards coupling reaction. In particular, N-heterocycles with directly connected N atoms such as tetrazine ring are supposed to be able to synergistically activate nitrogen and carbon sources to promote the formation of C-N coupling products. Corresponding description has been added into the revised version of manuscript.

2. *Figure 1b is not clear, which is suggested to be redrawn.*

Answer: Thanks a lot for this suggestion. Accordingly, **Figure 1b** has been modified in the revised version of manuscript.

3. *The bandgap calculated from Figure 2e is not consistent with those shown in Figure 2d.*

Answer: Thanks a lot for pointing out this error. The band gap in **Figure 2e** has been recalculated and corrected in the revised version of manuscript.

4. *The unit of the ordinate in Figure 2e should be "V vs. NHE" rather than "eV", while the unit of photocurrent was missing in Figure 2f.*

Answer: Thanks a lot for pointing out this error. Accordingly, "eV" has been changed into "V vs. NHE" in **Figure 2e** in the revised version of manuscript. In addition, the units of photocurrent have been added in **Figure 2f** in the revised version of manuscript.

5. *Some acronyms such as PXRD and UV-vis-DRS should be defined at their first*

usage.

Answer: Thanks a lot for this instruction. Accordingly, the acronyms including PXRD and UV-vis-DRS have been defined at their first usage in the revised version of manuscript.

6. The authors claimed “This is also supported by the almost unchanged PXRD pattern and XPS spectrum of 3D-TBBD-COF after photocatalytic cycles, Figures S30 and S31”. However, Figures S30 and S31 show that the spectra after cycling are not the same with from those before cycling. The authors should comment on this phenomenon.

Answer: Thanks a lot for this suggestion. In the PXRD pattern, the positions for the main peaks of 3D-TBBD-COF are not shifted before and after the photocatalytic cycle, implying the maintenance of the framework structure for 3D-TBBD-COF during the photocatalytic reaction. However, the signal-to-noise ratio of the PXRD pattern for the recycled sample slightly gets lowered due probably to the adsorption of solvent (H_2O) and reactant (CO_3^{2-} and NH_4^+) on the sample. This is also the reason for the slight difference in the XPS spectra of 3D-TBBD-COF before and after photocatalytic cycles. To confirm this point, the recycled 3D-TBBD-COF sample was further washed by deionized water for three times and dried at 60 °C in vacuum for 24 h, which was then used for the PXRD and XPS tests. As shown in **Figures S31 and S32** in the revised version of Supporting Information (actually **Figure R6** also given below), the recycled 3D-TBBD-COF sample exhibits very similar PXRD pattern and XPS spectrum to those for the 3D-TBBD-COF sample before photocatalytic cycles.

Figure R6. (a) PXRD patterns and (b) XPS N 1s spectra of 3D-TBBD-COF before and after photocatalytic reactions.

7. For comparison, the adsorption structures and adsorption energy for CO_2 , NH_3 , and CO_2+NH_3 on 3D-TPT-COF should be also provided.

Answer: Thanks a lot for this comment. Accordingly, the adsorption structures for CO_2 , NH_3 , and CO_2+NH_3 on 3D-TPT-COF have been added in **Figure S41** in the revised version of Supporting Information (actually **Figure R7** also given below). On

the basis of the adsorption energy calculations at the level of M06-2X/6-311G(d), the adsorption energy for NH₃, CO₂, and CO₂+NH₃ on 3D-TPT-COF amounts to -1, -17, and -11 kJ mol⁻¹, respectively. As can be found, the co-adsorption energy for CO₂+NH₃ on 3D-TPT-COF, -11 kJ mol⁻¹, is much lower than those on 3D-PDDP-COF (-87 kJ mol⁻¹) and 3D-TBBD-COF (-136 kJ mol⁻¹).

Figure R7. Adsorption structures and adsorption energies for NH₃, CO₂, and CO₂+NH₃ on 3D-TPT-COF.

Reviewer: 3

Comment: This is an interesting manuscript on the topic of Covalent Organic Frameworks for Urea Photosynthesis from NH₃ and CO₂. The manuscript is well-written, and the logic behind the work is sound. Therefore, I would like to recommend acceptance with minor revisions.

Answer: We truly appreciate the reviewer's affirmation regarding our work. The responses to the specific comments are shown below.

1. The COFs exhibit modest porosity. Is this the reason for the overall moderate performance? Additionally, the rationale behind selecting these specific COFs should be elaborated upon.

Answer: Thanks a lot for this comment and question. Indeed, the pore structure of the COFs-based photocatalysts including the pore size and pore volume affects the exposure of the active sites and mass transfer during photocatalytic reaction process, which in turn influences the overall photocatalytic performance. Actually it has been revealed that combination of square-planar and trigonal-planar building units is able to generate high symmetry 3D COFs with relatively high pore volume and relatively large pore size, see for examples in *Nat. Commun.* **2018**, *9*, 5274; *J. Am. Chem. Soc.* **2020**, *142*, 16346; and *J. Am. Chem. Soc.* **2020**, *142*, 20335. This is the reason that the three COFs are designed and constructed from square-planar and trigonal-planar building units *via* a [4+3] condensation reaction in the present work. The interconnected nanochannels and modest pore size of these COFs are favorable to the exposure of the active sites and mass transfer. In addition, these three

heterocyclic-N-containing COFs possessing the same topology and similar pore structure are considered to be suitable for comparative study to demonstrate the effect of the heterocyclic N active sites microenvironment on their catalytic performance towards photosynthesis of urea. Corresponding description has been added into the revised version of manuscript.

2. *Many COFs have been reported for use in urea synthesis. How does the performance of these COFs compare to those previously reported? Furthermore, the authors should provide details on the chemical stability of the COFs before and after catalysis.*

Answer: Thanks a lot for this instruction. It is noteworthy that thus far only one COF-based material has been revealed to show catalytic activity for the electrosynthesis of urea, in detail please see *Sci. China Chem.* **2023**, *66*, 1417–1424. To the best of our knowledge, COFs have not yet been employed for photocatalytic urea synthesis. Actually, as detailed in the Introduction section of the original manuscript, quite recently few trials over urea photosynthesis with N₂ and CO₂ as starting material and H₂O as hydrogen source catalyzed by inorganic nano-catalysts have been reported, in detail please see *Adv. Mater.* **2022**, *34*, 2207793; *Catal. Sci. Technol.* **2023**, *13*, 1855; *Angew. Chem. Int. Ed.* **2023**, *62*, e202312076; *Adv. Energy Mater.* **2024**, *14*, 2400201; and *Angew. Chem. Int. Ed.* **2024**, *63*, e202405637. However, the photocatalytic performance is unsatisfactory in terms of both the low urea production rate in the range of 6.4–133 μmol g⁻¹ h⁻¹ and in particular poor selectivity as clearly revealed in these reports. Nevertheless, it is worth noting that urea synthesis from N₂ and CO₂ starting materials and H₂O as hydrogen sources under either thermal, or electrochemical, or photochemical catalysis has to face the very high activation energy of 945 kJ mol⁻¹ for the triple N≡N bond and the very high reaction enthalpy change ($\Delta_r H_m^\ominus$) of 1264 kJ mol⁻¹ for 2N₂ + 2CO₂ + 4H₂O = 2CO(NH₂)₂ + 3O₂. In contrast, employment of NH₃ and CO₂ as starting materials on the basis of the present industrial urea production route, the C-N coupling reaction for the urea generation would become a thermodynamically spontaneous one with a negative $\Delta_r H_m^\ominus$ of -135.5 kJ mol⁻¹ for NH₃ + CO₂ = CO(NH₂)₂ + H₂O, in detail please see *Joule* **2024**, *8*, 1224-1238; *Nat. Chem.* **2020**, *12*, 717-724; and *Energy Environ. Sci.* **2012**, *5*, 8417-8429. Furthermore, in comparison with the stable N≡N bond in N₂, the lone pair electrons in NH₃ are naturally reactive center, in favor of fast reaction dynamics. As a total result, photosynthesis of urea from NH₃ and CO₂ appears more promising towards future practical industrial application. This, however, seems to remain still essentially unexplored, limited to the very lately trial over Pd@TiO₂/Gr catalyzed urea photosynthesis with a series of nitrogenous sources including NO₃⁻, N₂, and NH₃ as the sole report, in detail please see *CCS Chem.* **2024**, *6*, 3008–3017.

In the present case, three 3D COFs (3D-TPT-COF, 3D-PDDP-COF, and 3D-TBBD-COF) were designed with their urea photosynthesis performance from NH₃ and CO₂ investigated. Among which 3D-TBBD-COF displays a urea production

rate of $523 \mu\text{mol g}^{-1} \text{h}^{-1}$, significantly outperforming the reported photocatalysts. In addition, to prove the stability of 3D-TBBD-COF during photocatalytic reaction process, the PXRD patterns, XPS spectra, and FTIR spectra of 3D-TBBD-COF before and after photocatalytic cycles have been recorded and compared. As can be seen in **Figures S31-S33** in the revised version of Supporting Information (actually **Figure R8** also given below), the recycled 3D-TBBD-COF sample after photocatalytic cycles exhibits very similar PXRD pattern, XPS spectrum, and FT-IR spectrum to those for the 3D-TBBD-COF sample before photocatalytic cycles, revealing its excellent chemical stability during photocatalytic reaction process.

Figure R8. (a) PXRD patterns, (b) XPS N 1s spectra, and (C) FT-IR spectrum of 3D-TBBD-COF before and after photocatalytic reactions.